# Player Heart Rate Responses and Pony External Load Measures during 16-Goal Polo

**Russ Best** [1,2] 

[1] Centre for Sport Science and Human Performance, Waikato Institute of Technology, Hamilton 3288, New Zealand; Russell.Best@wintec.ac.nz

[2] School of Health and Social Care, Teesside University, Middlesbrough TS1 3BX, UK

**Abstract:** This dataset provides information pertaining to the spatiotemporal stresses experienced by Polo ponies in play and the cardiovascular responses to these demands by Polo players, during 16-goal Polo. Data were collected by player-worn GPS units and paired heart rate monitors, across a New Zealand Polo season. The dataset comprises observations from 160 chukkas of Open Polo, and is presented as per chukka per game (curated) and in per effort per player (raw) formats. Data for distance, speed, and high intensity metrics are presented and are further categorised into five equine-based speed zones, in accordance with previous literature. The purpose of this dataset is to provide a detailed quantification of the load experienced by Polo players and their ponies at the highest domestic performance level in New Zealand, as well as advancing the scope of previous Polo literature that has employed GPS or heart rate monitoring technologies. This dataset may be of interest to equine scientists and trainers, veterinary practitioners, and sports scientists. An exemplar template is provided to facilitate the adoption of this data collection approach by other practitioners.

**Keywords:** polo; GPS; equine; equid; heart rate; speed; distance; performance analysis; equestrian sport

## 1. Summary

Polo is an equestrian team sport played by two teams of four players, mounted on horseback [1]. Players must change horses between periods of play (chukkas), which are contested over 7–7.5 min of regulation play [1]. The actual duration of chukkas varies widely, with values up to 19:27 reported [2,3], and differences are also apparent between Open and Women's Polo [4]. This poses significant stress upon players' horses; hence, players may change horses during a chukka to manage a horse's workload across a game or tournament.

The level of Polo being played is denoted by the cumulative handicap of all players on a Polo team (e.g., 16 goals), with individual player handicaps being rated from −2 (low ability/beginner) to +10 (exceptional) goals [1]. Sixteen goal Polo is the highest level of Polo typically contested in New Zealand, with the exception of international matches [3], which are considered 'Open', where any cumulative handicap can be played. Whilst we have previously characterized the spatiotemporal demands of Polo from the 0–24 goal level [3], the metabolic cost of playing Polo is largely unexplored, with minimal work assessing players or ponies published to date [5–8]. These data and future datasets may inform sport-specific nutrition and recovery strategies, tactical decisions, and coaching strategies, and may facilitate the design of a Polo-specific fitness test. As per previous recommendations [9], this dataset provides a platform to build upon this previous work by allowing Polo player heart rate (internal load) to be contextualized against the work performed (external load) by their mounts, thus we can examine how players' physiology and ponies' performance interact, and how this may contribute to game outcome or be affected by environmental factors. Future investigations may wish to pair

other data with the (internal and external) load characteristics described below, to further quantify the responses to exercise upon a wide array of physiological systems e.g., equine or human blood analyses or temperature changes assessed through thermography or thermometry.

## 2. Data Description

The subheadings below outline categories of data that appear within the dataset; these data are presented on a per chukka and per game basis, and hence are considered to be averages of the samples derived from accelerometer, GPS, and magnetometer data. Accelerometer and magnetometer values are excluded to not obfuscate the reader and scientists who are most likely to use this dataset, and are likely unfamiliar with how to interpret data produced by these devices. We have previously shown the manufacturer's interpretation of these outputs, using their in-house software (VX Sport, VX Sport, Lower Hutt, New Zealand), to produce reliable results when the GPS device is worn, as specified in the methods section below [10], and have previously published a dataset under similar constraints [2]. Descriptive characteristics i.e., time of day and date, are also included within the dataset for the sake of completeness.

### 2.1. Player Heart Rate Metrics

Average (mean) and maximum heart rate were recorded for each chukka (trimmed period). Average heart rate is not reported on a per game level, as this would require the inclusion of enforced rest periods (e.g., half-time) in the dataset, and may artificially skew our understanding of the physiological demands of Polo by significantly lowering the average heart rate for a game, and thus increase the chances of committing a type two error. Heart rate data that appear to be inconsistent within an individual are not excluded from this dataset for the sake of completeness, but are presented in a raw edit of the dataset as supplementary material (S2). It is recommended that these values be excluded from future analyses as they are 'outliers' by definition of being physiologically implausible for the individual, based upon repeated observations of that individual under the same exercise scenario.

### 2.2. Speed Metrics

Maximum speed and average speed (both km/h) attained per chukka are reported for each player for each chukka; All subsequent categories that rely upon speed (2.3 distance metrics and 2.4 time metrics) are divided into five speed zones, which are derived from an assumed maximum speed of 60km/h, and these zones are as follows: Zone 1: 0–19.2 km/h; Zone 2: 19.2–23.4 km/h; Zone 3: 23.4–28.2 km/h; Zone 4: 28.2–47.4 km/h; Zone 5: 47.4–60 km/h. These zones have been used repeatedly in our previous work across levels of Polo [2,3,10] and between Open and Women's Polo [4], and approximate to horse gait speeds of walk/trot, canter, fast canter–gallop, gallop, and maximal effort [11], although it should be noted that equine gaits tend to be categorized via footfall patterns, as opposed to velocity [4,11,12].

### 2.3. Distance Metrics

Total distance per chukka is reported (m) for each player. Distance covered in each speed zone (as per 2.2 speed metrics) is also reported to provide a more in-depth understanding of the external load experienced by each players' horses. Notational analysis accompanied these games and so can be used to note the points at which horses were changed by each player; taken together, this provides an understanding of how far and fast a horse went whilst they were on the field. Relative distances per chukka (m/min) are also reported, as this parameter has been used across other team sports as a proxy for game intensity, in the absence of heart rate data [13].

*2.4. Time Metrics*

Time in each of the five speed zones is reported, as is the duration of each chukka (min:sec). Chukka length is considered the 'real' time difference between the start and end of each chukka, which will always consist of between 7:00 and 7:30 of regulation Polo play.

*2.5. High Intensity Metrics*

High intensity metrics include sprints, accelerations, decelerations, and impacts, and their associated parameters. Sprints are considered a positive or negative acceleration > 3 m/s/s [3,4,10] and are described as a count variable on a per chukka basis for game outcome analyses. Accelerations and decelerations are also considered on a count basis, with descriptive data pertaining to average length, maximum length, and total length of each effort (m) provided. Further descriptive data pertaining to starting, maximum and average velocity, and duration of each effort accompany accelerations and deceleration counts, and by extension sprints in the raw data for each game. Impacts are gathered through changes in accelerometer measurements and are provided as a count variable, on a per chukka basis, and are derived as per the manufacturer's proprietary algorithm/software (VX Sport, VX Sport, Lower Hutt, Wellington).

## 3. Methods

The investigations were carried out following the rules of the Declaration of Helsinki of 1975 (http://www.wma.net/en/30publications/10policies/b3/), revised in 2008, and the International Guiding Principles for Biomedical Research Involving Animals, as issued by the Council for the International Organizations of Medical Sciences. Ethical approval for this research was provided by the Waikato Institute of Technology's Human Ethics Research Group (ID: WTFE2601102018) and Animal Ethics Committee (no identification number provided). Players who participated in this project held a current New Zealand Polo Association handicap of 1–7 goals, and had previously played at the 16-goal level, either in New Zealand or elsewhere. Players' handicaps are removed to preserve player and team anonymity.

Data were collected using GPS devices (VX Sport 350, VX Sport, Lower Hutt, New Zealand) and heart rate monitors (Suunto Smart Sensor, Suunto, Vantaa, Finland), sampling and transmitting at 10 Hz and 2.4 Ghz, respectively. GPS devices were set to an equestrian mode with a customized range of 0–60 km/h as previously reported [2–4,10]. It is important to note that this does not limit recording ≤ 60 km/h, but sets a benchmark in the accompanying analysis software, for which, to scale the data, collected GPS devices were turned on upon arrival at the ground, to secure an initial 'satellite lock', and left to sample for 1–2 min, effectively registering the location for the sample, before being turned off until game sampling. Units were turned on ~45 min prior to games starting, and placed inside customized pouches, secured with electrical tape to prevent undue oscillation of the device during play. GPS devices were affixed to players' belts in their customized pouches (as per Figure 1), with the corresponding paired heart rate monitors worn around players' chests, with sampling electrodes positioned either side of a player's sternum. Equipment was delivered to players ~30 min prior to games starting, to allow players to warm-up and undertake any preparatory routines without experimenter interference.

Following the game devices were collected, and data downloaded using the manufacturer's software (VX Sport, VX Sport, Lower Hutt, New Zealand). Data files for each player were then 'trimmed' to game duration and divided or 'split' into chukkas (chukka duration derived from accompanying notational analysis) using the software's Trim and Split function. These data were then exported to Microsoft Excel (Microsoft Excel for Mac, Version 15.30, Microsoft Corporation, Redmond, WA, USA) and converted into a match report for each player. Match reports were limited to durations and distances covered in each speed zone, the number of impacts per chukka, average and top speeds attained per chukka, and player mean and maximum heart rate per chukka.

All data, including an exemplar match report, are provided as a spreadsheet in supplementary materials; these data are presented on a per game basis, initially as per chukka values, and subsequently as raw data (i.e., per 'effort', per player).

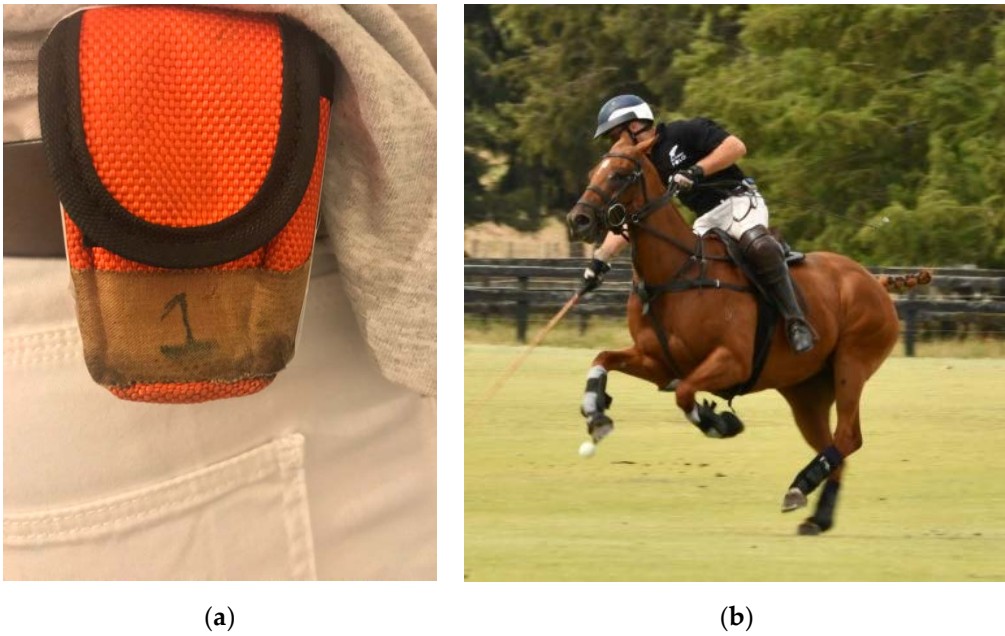

(**a**) (**b**)

**Figure 1.** (**a**) belt mounted GPS unit in close-up; (**b**) belt mounted GPS unit on a Polo player, during match-play.

## 4. User Notes

An exemplar player report has been attached as part of the dataset/supplementary materials. This report details the breakdown of distance covered per speed zone within each chukka, and is provided to each player within a team. This is simply one way of presenting these data, and other options are currently being explored that better detail the interaction between players. For example, all players' data plotted per chukka against team averages for each chukka, for characteristics appropriate to the research question. All player or team identifiers have been removed from this exemplar.

Equine scientists can use these data to gain an understanding of the external load experienced by Polo ponies, per chukka and per game, and how these may differ between games too. The play of an extra seventh chukka for instance, is an unusual but welcome addition to the dataset, as players will manage their strings for six chukkas, and so the pony played in the seventh chukka has likely already completed a significant volume of work prior to being played in a decisive period. This may or may not increase the risk of injury, or impose an extra recovery cost upon that pony. Sport and exercise scientists can use these data to gain an appreciation of work:rest ratios experienced during 16-goal Polo play, but more data are required across a range of levels of play to ascertain how level of play, playing position, and a player's handicap interact and manifest in their cardiovascular responses to the work performed by their ponies, and how players manage their own physiological fatigue with that of their mounts.

**Supplementary Materials:** The data described in this manuscript are available as supplementary material. The following are available online at http://www.mdpi.com/2306-5729/5/2/34/s1. S1 presents a clean dataset, and S2 presents a raw dataset.

**Funding:** This research received no external funding.

**Acknowledgments:** The author would like Regan Standing for independent support during the project, and Nik Whitfield for providing technical assistance prior to data collection taking place. The author would also like to thank the players who generously donated their time to the project across the high goal season and tournaments therein, and finally Bodian Photography for the kind provision of the photograph in Figure 1, panel B.

**Conflicts of Interest:** The author declares no conflict of interest.

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
