# Peer review of "Player Heart Rate Responses and Pony External Load Measures during 16-Goal Polo"

_data_

Round 1
Reviewer 1 Report
Good to see research being undertaken in polo, which is often a neglected equestrian discipline. The methods outlined and data presented have the potential to support future research in this field.
The author’s guidelines for the journal suggest Data Descriptors require the information presented (which aligns to the template) in addition to the sections included in a research manuscript submission – materials and methods, results, discussion and conclusions are not provided.
Abstract:
Good overview of data collected, covering purpose and potential audience but this should also include a summary of key results and conclusions based upon them.
Line 10: please insert in after play
Dataset and dataset licence: sections require completion.
Keywords: suggest adding in performance analysis and equestrian sport
Summary:
Clear rationale for study provided with sufficient background to polo included to support readers less familiar with the sport.
Line 32: values omitted (X within text)
Line 37: would be worthwhile to define how goal rating aligns to performance for the less familiar reader (e.g. -2 (low ability) or via a footnote)
Line 39: suggest replacing meaning with where
Data description:
Generally clear and informative data descriptions provided
Line 68 – 72: I would question retaining individuals with inconsistent HR data in the dataset presented as you are recommending these should be excluded for future studies, as this feels counterintuitive. Could you present a ‘clean’ dataset (as this would be more valuable as a baseline evidence base for future research) and then the additional data as a sub-set?
Methods:
Detailed description of methods used provided
Line 122: suggest removing ‘-‘ and starting a new sentence from It is important…
Line 137: please insert version and manufacturer’s details for Excel
User notes: would be good to suggest / outline the other methods which could be used to interpret the data
Supplementary materials: These should be presented as Appendix A as per journal guidelines – including units and explanations for all symbols used within datasets would be beneficial. Including further guidance on how to interpret data from an applied perspective which could be utilised by relevant industry professionals would be useful to include in the supplementary materials.
Author Response
I thank the reviewer for a prompt review, with clear guidance as to how the manuscript and accompanying dataset are to be improved.
Please find below a point by point response to the reviewer's comments:
Good to see research being undertaken in polo, which is often a neglected equestrian discipline. The methods outlined and data presented have the potential to support future research in this field.
Thank you, this is part of ongoing work in Polo for our research group and it certainly is an exciting but challenging area of research that appears to be growing, globally.
The author’s guidelines for the journal suggest Data Descriptors require the information presented (which aligns to the template) in addition to the sections included in a research manuscript submission – materials and methods, results, discussion and conclusions are not provided.
As per the template, the recommended sections are not included. I'm unsure as to where this guidance is provided in author guidelines, as in the manuscript preparation section, under the subsection general guidance the description of a data descriptor does not include these sections (materials and methods, results, discussion and conclusions), nor are they included in the data descriptor information subsection.
Abstract:
Good overview of data collected, covering purpose and potential audience but this should also include a summary of key results and conclusions based upon them.
Amended to include key elements of the dataset
Line 10: please insert in after play
Dataset and dataset licence: sections require completion.
Keywords: suggest adding in performance analysis and equestrian sport
All amended as requested
Summary:
Clear rationale for study provided with sufficient background to polo included to support readers less familiar with the sport.
Line 32: values omitted (X within text)
Line 37: would be worthwhile to define how goal rating aligns to performance for the less familiar reader (e.g. -2 (low ability) or via a footnote)
Line 39: suggest replacing meaning with where
All amended as requested
Data description:
Generally clear and informative data descriptions provided
Line 68 – 72: I would question retaining individuals with inconsistent HR data in the dataset presented as you are recommending these should be excluded for future studies, as this feels counterintuitive. Could you present a ‘clean’ dataset (as this would be more valuable as a baseline evidence base for future research) and then the additional data as a sub-set?
A clean dataset is now provided as S1 and S2 is inclusive of the excluded data
Methods:
Detailed description of methods used provided
Line 122: suggest removing ‘-‘ and starting a new sentence from It is important…
Line 137: please insert version and manufacturer’s details for Excel
All amended as requested
User notes: would be good to suggest / outline the other methods which could be used to interpret the data
Amended as requested - lines 149 - 153 now read:
'This is simply one way of presenting these data, and other options are currently being explored that better detail the interaction between players. For example all players’ data plotted per chukka against team averages for each chukka, for characteristics appropriate to the research question. All player or team identifiers have been removed from this exemplar.'
Supplementary materials: These should be presented as Appendix A as per journal guidelines – including units and explanations for all symbols used within datasets would be beneficial. Including further guidance on how to interpret data from an applied perspective which could be utilised by relevant industry professionals would be useful to include in the supplementary materials.
With respect to presentation of these as Appendix A, I cannot find author guidelines to this effect but have followed the recommendation of the template that suggests if data are to be uploaded as supplementary materials as opposed to independently housed then they be appended as per the following 'If the dataset is submitted and will be published as a supplement to this paper in the journal Data, this field will be filled by the editors of the journal. In this case, please make sure to submit the dataset as a supplement when entering your manuscript into our manuscript editorial system.'
Further guidance has been included for equine and sport and exercise scientists, as suggested; this has been included in the user notes (lines 154 - 164) as well as in the data document itself via comments on specific cells. Comments have also been used to describe variable headings, as suggested.
Thank you again for a thorough review, I feel the data descriptor has been strengthen and thus more easily analysed and applied to the field as a result.
Reviewer 2 Report
Dear Authors,
Thank you for submitting this report. I am quite sure that many researchers need such data to understand more productive research that focuses on this type of parameter and this sport.
I have no specific comments and I would accept the report as it is.
Regards
Author Response
Thank you for the kind review and opportunity to present the work as it currently stands